# NF-κB Signaling in Ovarian Cancer

**DOI:** 10.3390/cancers11081182

**Published:** 2019-08-15

**Authors:** Brittney S. Harrington, Christina M. Annunziata

**Affiliations:** Women’s Malignancies Branch, National Cancer Institute, Bethesda, MD 20892, USA

**Keywords:** nuclear factor-kappa B, RelA, RelB, stemness, chemoresistance, ovarian cancer

## Abstract

The NF-κB signaling pathway is a master and commander in ovarian cancer (OC) that promotes chemoresistance, cancer stem cell maintenance, metastasis and immune evasion. Many signaling pathways are dysregulated in OC and can activate NF-κB signaling through canonical or non-canonical pathways which have both overlapping and distinct roles in tumor progression. The activation of canonical NF-κB signaling has been well established for anti-apoptotic and immunomodulatory functions in response to the tumor microenvironment and the non-canonical pathway in cancer stem cell maintenance and tumor re-initiation. NF-κB activity in OC cells helps to create an immune-evasive environment and to attract infiltrating immune cells with tumor-promoting phenotypes, which in turn, drive constitutive NF-κB activation in OC cells to promote cell survival and metastasis. For these reasons, NF-κB is an attractive target in OC, but current strategies are limited and broad inhibition of this major signaling pathway in normal physiological and immunological functions may produce unwanted side effects. There are some promising pre-clinical outcomes from developing research to target and inhibit NF-κB only in the tumor-reinitiating cancer cell population of OC and concurrently activate canonical NF-κB signaling in immune cells to promote anti-tumor immunity.

## 1. Introduction

Ovarian cancer (OC) is the most lethal gynecologic malignancy [1]. The low survival rate of approximately 40% owes to the intra- and inter-tumor heterogeneity, diagnosis at an advanced stage and high rate of recurrence and chemoresistance that classifies this disease [2]. There is increasing evidence that the high rate of disease recurrence after chemotherapy is due to residual populations of cancer stem cells (CSCs) that escaped therapy and re-establish the disease, leading to progression [3]. Populations of CSCs have been identified by certain markers in OC such as CD133, CD44, and ALDH and efforts are focused on strategies to target these populations to prevent recurrent disease [3,4]. The inflammatory but immunosuppressive tumor microenvironment also contributes to OC progression by the production of cytokines that promote survival signaling in OC cells and suppress the infiltration of anti-tumor immune cells [5].

The increased expression of NF-κB signaling molecules had been associated with poor prognosis in many cancers including bladder cancer [6], non-small-cell lung cancer [7] and renal cell carcinoma [8,9]. In OC, the expression levels of NF-κB proteins, p65 and p50, are elevated in malignant and borderline serous ovarian tumors [10,11] and the elevated expression of subunits of the NF-κB pathway, such as the activating IκB kinases (IKKs) β and ε, have been associated with worse overall survival in OC [12,13,14,15]. Constitutive activation of NF-κB was found to be a feature of cell lines derived from granulosa cell tumors of the ovary, thought to be driven by IKKβ; however, the upstream drivers of this pathway were not fully elucidated [16]. In terms of prognostic value, the elevated expression of NF-κB subunits, p65 and p50, has been reported as being greater than 90% of OC cases examined [17,18] and is, therefore, not useful to distinguish patients based on expression of these markers alone. Correlative studies investigating NF-κB pathway proteins and other markers have been reported. The inhibitory proteins that sequester NF-κB in the cytoplasm IκB inversely correlate with p65 expression in OC with poor prognosis, where loss of IκB and increased total p65 expression was associated with worse overall survival [19]. Similarly, the high expression of NF-κB p65 combined with low expression of tumor suppressor PTEN were found to be independent risk factors for chemoresistant ovarian cancer [18]. Another study, focusing on upstream activators of canonical NF-κB signaling in OC, found that high expression of both toll-like receptor 4 (TLR4) and Myd-88 was associated with a poorer outcome and a more aggressive tumor phenotype and was significantly correlated with the expression of NF-κB pathway proteins IKKβ and matrix metalloprotease 9 (MMP9) [17]. The prognostic significance of NF-κB activity in OC is related to its roles in promoting properties of aggressive OC such as chemoresistance [15,20,21,22,23,24] and maintenance of cancer stem cell populations that contribute to recurrence [3,25].

The NF-κB signaling pathway has been implicated in the progression of many cancers, in both blood and solid malignancies [26]. The NF-κB pathway controls multiple cellular processes that can be exploited by cancer cells such as the inhibition of apoptosis in response to chemotherapy, transcription of self-renewal genes in CSCs and the production of anti-tumor cytokines for immune evasion [27]. Due to its role in many normal cellular functions, targeting NF-κB in tumor cells presents a challenge in OC [28]. In this review, we examine the role of the NF-κB signaling pathway in OC chemo-resistance and disease recurrence, inflammation and immunosuppression, and the strategies for targeting this pathway clinically in OC.

## 2. Canonical and Non-Canonical NF-κB Signaling in OC

The activation of NF-κB signaling in OC has been shown to promote invasiveness, stemness and chemoresistance. NF-κB is a group of transcription factors consisting of five proteins; RelA/p65, c-Rel, RelB, p50 (NF-κB 1) and p52 (NF-κB 2), which are generated by selective degradation in the proteasome of p105 and p100, respectively [28,29]. The composition of the subunits driving NF-κB transcriptional activity directs the signaling towards the canonical or non-canonical pathway. The canonical pathway activates c-Rel-p50 or RelA-p50/p52, while the non-canonical pathway typically activates RelB-p52 [28,29]. Under basal conditions when NF-κB signaling is inactive, the subunits are sequestered in the cytoplasm bound to inhibitory proteins that must be degraded to allow nuclear translocation for transcriptional activity. Inactive c-Rel or RelA/p65 is bound to an inhibitor of κB (IκB) or p105, both of which can be proteasomally degraded after phosphorylation by IκB kinases (IKK) to generate the active RelA/p50 dimer for nuclear translocation and activation of canonical signaling [29,30]. Similarly, RelB is bound to p100 in an inactive state, which can also be phosphorylated by an IKK, but this results in ubiquitination and partial degradation of p100 which generates p52 in the proteasome for nuclear translocation and non-canonical transcriptional activity [29,30].

There are overlapping and distinct activators of the canonical and non-canonical NF-κB signaling pathways and these can differ depending on the cell type [28]. Proinflammatory signaling can activate canonical NF-κB signaling in the innate immune response to pathogens and mitogen-activated protein kinase (MAPK) signaling can enhance cell survival through cellular inhibitors of apoptosis proteins (cIAPs) and canonical NF-κB signaling [29]. Non-canonical NF-κB signaling can be activated in response to adaptive immunity, for B cell activation and expansion and in bone remodeling [31]. Recently, research efforts have focused on delineation of the canonical and non-canonical pathways for their roles in disease progression.

### 2.1. Canonical NF-κB Signaling in Proliferation and Chemoresistance

The canonical pathway of NF-κB activation has been well established for its anti-apoptotic, pro-angiogenic function in many cancers [32,33]. NF-κB transcription is activated after the degradation of inhibitory IκB proteins by phosphorylation by IKKs, allowing translocation to the nucleus with either RelA/p65 (canonical) or RelB (non-canonical) signaling pathway subunits [25,27,34]. Due to their critical function in the activation of NF-κB signaling, IKKs have been investigated as a target for inhibitors to prevent the nuclear translocation of NF-κB subunits for transcriptional activity. In OC, IKKβ was shown to promote the anchorage-independent growth, proliferation and invasion of OC cells [13]. The pro-angiogenic chemokines IL-8 and VEGF are direct targets of NF-κB and are upregulated by activation of canonical NF-κB signaling through IKKs [13,35,36].

Canonical NF-κB signaling through p65/RelA can be activated in response to inflammatory chemokines and cytokines in the tumor microenvironment. Increasing M2 (pro-tumor) macrophage infiltration and increased p65 phosphorylation and NF-κB activity during tumor progression could be observed intra-vitally in a syngeneic mouse model of OC, using an NF-κB-dependent GFP/luciferase reporter cell line [37]. Inhibition of NF-κB activity in this model was associated with decreased expression of M2 markers and increasing expression of M1 (anti-tumor) markers [37]. This has been confirmed in vitro, where increased expression of p65 is observed in OC cells co-cultured with macrophages, which also showed increased invasiveness and migration compared to OC cells alone [38,39]. Inhibition of NF-κB signaling by inhibitors or RNA interference significantly reduced the invasive phenotype of the co-cultured OC cells [39].

NF-κB activity mediates resistance to stressors of OC cells that occurs during disease progression and metastasis such as oxidative stress and anchorage-independent survival [40]. Inhibition of NF-κB activity in cultured OC cell lines significantly inhibited anchorage-independent growth [40]. Canonical NF-κB signaling regulates the expression of several antioxidant response genes to alleviate elevated cellular reactive oxygen species levels in OC cells [40]. Similarly, numerous studies have shown that the NF-κB pathway is activated in platinum-resistant OC cells to modulate apoptosis and promote cell survival [20,22,41,42,43]. In one example, OC cell lines with resistance to multiple therapies (platinum, paclitaxel and erlotinib) had high NF-κB activity, as measured by p65 phosphorylation, which increased with increasing doses of chemotherapy drugs [21]. Treatment with the NF-κB inhibitor BAY 11-7082 significantly reduced the viability, clonogenicity and anoikis resistance of OC cell lines as a monotherapy in vitro and sensitized cells to multiple therapies [21]. The proposed mechanism of action of BAY 11-7082 appeared to be inhibition of anti-apoptotic genes cIAP1, Bcl-2, and Bcl-xl, direct targets of RelA, and enhanced the expression of p21, a regulator of cell proliferation in the resistant cell lines [21].

DNA damage can also activate NF-κB signaling in a retrograde nuclear to cytoplasmic signaling cascade that is dependent on ataxia telangiectasia-mutated (ATM) [44]. Chemotherapeutic agents that cause genotoxic stress, such as etoposide and doxorubicin, activate NF-κB pro-survival signaling [45,46]. In response to DNA damage, the IKKγ subunit—also known as NF-κB essential modulator (NEMO)—translocates into the nucleus for post-translational modification, phosphorylation by ATM and monoubiquitination for nuclear export [44,47]. In the cytoplasm, the modified NEMO-ATM complex can activate IKKs to phosphorylate and degrade inhibitory IκBα subunits and activate canonical NF-κB survival signaling [44].

Both pro- and anti-apoptotic signaling can be activated downstream of NF-κB in OC cell lines, suggesting a biphasic role for NF-κB in OC progression [24]. In a comparison of chemoresistant cell lines with high NF-κB transcriptional activity with their isogenic, chemosensitive parental cells, it was found that inhibition of the NF-κB pathway using a dominant negative phosphorylation-deficient mutant of IκBα had opposing effects on the chemoresistant and chemosensitive cells of the same origin [24]. The IκBα mutation caused reduced tumor growth of the aggressive, chemoresistant cells in vivo but this same mutation acted to promote tumor growth in the isogenic chemosensitive cells [24]. Apoptosis could be induced in the aggressive chemoresistant cell lines by introduction of this mutation, whereas the chemosensitive cells showed reduced apoptosis [41]. This polar difference in activity and cell viability was related to the ability of NF-κB to differentially regulate mitogen-activated protein kinase (MAPK) phosphorylation to produce pro- or anti- apoptotic signaling in OC cells [24]. This study highlighted the canonical pathway activity and its potential to contribute to chemoresistance and OC cell survival.

Canonical NF-κB signaling may be important in maintaining the proliferative cell populations of tumors, whereas non-canonical NF-κB signaling maintains the CSC populations. In a comparative study of shRNAs targeting RelA or RelB, OC cells grown in tumor-initiating cell (TIC) culture conditions had significantly fewer Ki67 positive, proliferative cell populations and reduced viability with loss of RelA, compared with the loss of RelB [4].

### 2.2. Non-Canonical NF-κB Signaling and OC Persistence

Activation of the non-canonical NF-κB signaling pathway has been associated with CSC maintenance and self-renewal in OC [4]. A number of genes have been identified as markers of CSCs in OC including CD133, CD117, aldehyde dehydrogenases (ALDH) and CD44 [48,49]. In CD44+ SKOV3 cells there was increased expression of RelA but the expression of non-canonical NF-κB proteins RelB and IKKα was greater, as was the expression of stem cell markers *NANOG*, *SOX2* and *OCT4/3* compared to CD44− cells [25]. The CD44+ population showed greater colony-forming capacity in vitro and tumorigenicity in vivo which could be reversed by knocking down IKKα or by expressing a dominant negative mutant in the CD44+ population [25]. In other cancer types, IKKα can be activated and function separately from NF-κB pathway signaling [50,51], but no direct evidence of this has been shown in OC as yet. Thus, the importance of IKKα in this model suggests that the CD44+ CSC population was likely maintained by the non-canonical NF-κB signaling pathway via RelB.

Similar results were reported in the CSC population in other OC cell lines, where both RelA and RelB expression increased in OC cells grown under TIC culture conditions but RelB was more highly expressed in the ALDH+/CD133+ CSC population of OV90 and ACI23 cell lines compared to RelA [4]. Depletion of RelA or RelB in cell line xenografts had differing effects based on the site of tumorigenesis. In subcutaneous and intrabursal xenografts of OC cell lines with inducible knockdown of RelA or RelB with shRNA, loss of either RelA or RelB significantly reduced tumor burden compared to controls [4]. However, intraperitoneal xenografts of shRelA and shRelB cells showed dramatically fewer numbers of tumor cells with shRelB compared to shRelA and control, suggesting there is a greater requirement of non-canonical NF-κB signaling in intraperitoneal, anchorage-independent metastasis in this model [4].

Further evidence of differing roles for canonical and non-canonical NF-κB signaling in OC progression was demonstrated in the context of ECM-interacting protein Transglutaminase (TG2). TG2 can promote OC metastasis in vitro and in vivo and can activate both canonical and non-canonical NF-κB signaling pathways [52,53]. TG2 signaling intracellularly activates canonical NF-κB and a secreted form of TG2 can activate non-canonical NF-κB [52]. In the latter case, RelB activation by secreted TG2 induced CD44 expression, downregulated E-cadherin, and increased invasiveness and peritoneal metastasis in vivo, but interestingly, did not affect cell proliferation in vitro [52]. This is consistent with previous findings about the functions of RelB in OC proliferative and CSC populations [4].

Non-canonical NF-κB signaling is also important for CSC maintenance in endometrioid endometrial cancer (EEC). RelB but not RelA, was significantly upregulated in EEC, and silencing RelB in EEC cell lines affected colony formation, tumor growth in vivo and cell cycle progression [54]. The cells silenced for RelB accumulated in G0/G1, upregulated cell cycle arresting proteins p27 and p21, and activated apoptosis [46]. In comparison, cells with enhanced RelB expression showed advanced cell cycle progression and upregulation of proliferative, apoptosis and cell-cycle-transition-regulating genes such as c-Myc, cyclins D1 and E, and Bcl-2 and Bcl-xl [54].

### 2.3. NF-κB in Promoting Stemness

Both canonical and non-canonical NF-κB pathways can contribute to OC stem-like properties. CSCs are defined by their capacity for tumor-initiation, self-renewal, resistance to therapy and contribution to cancer recurrence [55]. CD44-expressing OC cells (CD44+) had greater clonogenicity and increased RelA, RelB and IKKα expression compared with the CD44-negative population [25]. OC stem cell populations defined by CD44+ and Myeloid differentiation primary response 88 (Myd-88) expression have demonstrated self-renewal, chemoresistance and tumor-initiating properties and constitutive NF-κB activity [56]. CD44+ Myd-88+ OC populations further perpetuated a pro-inflammatory microenvironment that promoted a wound repair phenotype driven by cytokines that are targets of NF-κB, including IL-6, IL-8, MCP-1 and GROα [57]. The wound repair phenotype and expression of self-renewal genes such as *NANOG* and *SOX2* was reduced by treatment of the CD44+ Myd-88+ cells with BAY 11-7082, an inhibitor with broad activity against the proteasome and ubiquitin system used to inhibit IKK, preventing phosphorylation and degradation of IκBα [57]. This implicates NF-κB in the maintenance of ovarian CSCs and the proinflammatory microenvironment that supports their survival.

ALDH is another CSC marker reported in numerous solid cancers to promote sphere formation and self-renewal, as well as drug resistance by clearance of reactive oxygen species [58]. The expression of both RelA/p65 and RelB were increased in cells grown under a tumor-initiating cell (TIC) culture and both were required for sphere formation in vitro [4]. In this context, ALDH activity was a reliable marker of CSCs and cells grown in TIC cultures and selected for ALDH activity (ALDH+), and this population had greater chemotherapy resistance compared to ALDH− TIC cultures [4]. Furthermore, ALDH+ was significantly reduced in vitro and in vivo in cells silenced for RelB with inducible shRNAs, further demonstrating a role for both canonical and non-canonical NF-κB signaling in CSC maintenance in OC [4].

## 3. NF-κB Activity in the Inflammatory Ovarian Cancer Microenvironment

The OC microenvironment is unique in that multiple cell types other than the tumor cells are coopted to promote metastasis and inhibit immune activity [5]. There are both inflammatory and immunosuppressive components of the milieu that are controlled by NF-κB activity in the cancer cells, as well as infiltrating immune cells. OC cells secrete immunosuppressive cytokines to promote pro-tumor phenotypes in tumor-associated macrophages (TAMs), which in turn, secrete immunoinhibitory cytokines such as IL-10 and contribute to constitutive NF-κB activation in OC cells by secreting TNFα, promoting survival, CSC expansion and metastasis [59,60].

### 3.1. Modulation of NF-κB Activity in Infiltrating Immune Cells

TAMs are typically polarized in the pro-tumor ‘M2′ subtype which secrete high levels of IL-10 and TNFα, and decreased IL-12 to promote the immunosuppressive environment and increase invasiveness in cancer cells [61,62]. In a co-culture experiment of mouse bone-marrow-derived macrophages (BMDMs) with mouse ovarian cancer cell line ID-8, the importance of upstream activators of canonical NF-κB signaling was demonstrated by specific knockouts of Myd-88, IL1R, TLR2 and TLR4 in the BMDMs, which caused decreased invasion of ID-8 cells in vitro and decreased tumor growth in vivo [62]. Specific inhibition of the NF-κB classical pathway activating kinase IKKβ in BMDMs caused a shift in the polarization from the pro-tumor M2 subtype to anti-tumor M1 subtype in vitro and in vivo [62]. IKKβ-inhibited BMDMs had decreased production of immunosuppressive cytokines and increased nitric oxide synthesis to promote apoptosis and increased IL-12 production for cytotoxic immune activation [62].

The OC microenvironment contains both pro-inflammatory and immunosuppressive cells so there is a contribution of cytokines by both anti-tumor M1- and pro-tumor M2-polarized macrophages. While M2 macrophages may be recruited locally to the tumor cells to be immunosuppressive, M1 macrophages are also recruited and promote chronic inflammation and high levels of cytokines that are intended to promote cytotoxic T cell recruitment, but these can also act on OC cells to promote survival and metastasis [60,63,64]. For example, treatment with the conditioned medium (CM) of M1-polarized macrophages caused nuclear translocation RelA/p65 and activation of NF-κB transcription, increased invasion and migration in OC cell lines [65]. It was determined that the CM produced by M1-polarized macrophages contained high levels of TNFα and IL-6 and that TNFα activated NF-κB in the OC cells [65].

Dendritic cells are also able to infiltrate the OC microenvironment and can be switched from immunostimulatory to immunosuppressive [66]. A target for cancer immunotherapy is programmed cell death-1 (PD-1), which is a negative regulator of immune cytotoxicity that is expressed on immune cells, particularly T cells [67]. Regulation of immune cytotoxicity through PD-1 is directed by interaction with its ligand (PD-L1), which is expressed on normal cells but has also been detected on cancer cells as a strategy to avoid immune cytotoxicity [67]. DCs from OC tumors and ascites that expressed PD-1 had decreased anti-tumor activity due to inhibition of canonical, immunogenic NF-κB activity [68]. Blockade of PD-1 on the DCs restored their pro-inflammatory functions by activation of canonical NF-κB with degradation of IκBα, allowing the nuclear translocation of RelA/p65 [68]. In the PD-1-blocked DCs, the upregulation of co-stimulatory molecules and secretion of immunogenic and proinflammatory cytokines TNFα, Il-6 and G-CSF was found to be dependent on NF-κB [68]. Unfortunately, the co-activation of NF-κB activation in OC cells in response to these cytokines can be utilized to further promote the immunosuppressive pro-tumor environment.

### 3.2. NF-κB Activation in Ovarian Cancer Cells to Alter the Microenvironment

OC cells contribute to the maintenance of the tumor microenvironment by secreting immunosuppessive cytokines to prevent migration of anti-tumor immune cells and to polarize macrophages to a pro-tumor M2 subtype [61,69]. This phenomenon triggers the expression of multiple cytokines including TNFα from tumor-infiltrating macrophages exposed to OC cells [61]. Accordingly, constitutive NF-κB signaling is activated in OC cells in response to co-culture with M2 macrophages and in response to high TNFα [38,69]. In addition, constitutively activated NF-κB signaling in OC cells also leads to increased immunosuppressive cytokines to further promote tumor growth [70]. The activation of canonical NF-κB signaling in OC cells is a major component of the immunosuppressive cytokine secretion, and inhibition of the pathway in OC cells caused the immune cell infiltrates to switch to an anti-tumor phenotype in vitro and in vivo [70,71,72].

Persistent inflammation has been associated with ovarian cancer development, in women with pelvic inflammatory disease or in response to pathogenic infections of *Chlamydia trachomatis* or *Mycoplasma genitalium* [73]. Recognition of pathogen-associated molecular patterns (PAMPs) by cell receptors such as TLRs initiates the innate immune response by activating NF-κB, which stimulates the production of pro-inflammatory cytokines including TNFα, IL-6, IL-8, and MCP-1 [74]. Persistent, chronic inflammation following infection and constitutive NF-κB activity may contribute to ovarian carcinogenesis by producing pro-inflammatory and pro-angiogenic cytokines and through anti-apoptotic signaling [73].

Another way OC cells use NF-κB signaling for immune evasion is through PD-L1 upregulation [75]. In response to chemotherapy, the expression of p65/RelA and PD-L1 increased in OC cell lines in vitro and in vivo [75]. Silencing p65/RelA prevented PD-L1 upregulation, confirming its dependence on NF-κB activity [75]. Thus, the OC microenvironment activates NF-κB signaling in a variety of cell types to promote tumor cell survival, immune evasion and metastasis.

## 4. Strategies to Target NF-κB in Ovarian Cancer

Many signaling pathways are dysregulated in OC and are able to activate NF-κB signaling to promote chemoresistance, metastasis and cancer cell survival. Although NF-κB is an attractive target for therapy, it also presents a number of challenges due to its plethora of functions in diverse cell types (Figure 1). It is activated by multiple signaling pathways under both normal physiological and immunological conditions, as well as pathogenic cancer-promoting conditions [34]. As such, broad inhibition of the pathway can lead to unwanted side effects [28]. For example, the only clinically available NF-κB inhibitors are the proteasome inhibitors bortezomib and carfilzomib, that prevent degradation of the inhibitory IκB proteins. However, non-specific proteasome blockade serves to affect multiple pathways in both normal cells, as well as target cells [28,76]. Targeting NF-κB transcription factors themselves is difficult to access with drugs and upstream, NF-κB activation can occur by multiple oncogenic pathways, presenting further challenges but also perhaps the opportunity to specifically inhibit it in OC [28]. Indeed, strategies to target the critical activators of the pathway have led to some promising pre-clinical outcomes.

### 4.1. IKK Inhibition

The activation of NF-κB is critically dependent on IKKs phosphorylating the inhibitory subunits that sequester NF-κB in the cytoplasm and, therefore, these kinases represent an attractive target to inhibit NF-κB [77,78]. One of the most widely used NF-κB inhibitors experimentally is BAY 11-7082, which is described as an IKK inhibitor, although it has been shown to affect a variety of cellular targets [79]. It effectively inhibits NF-κB activity and nuclear translocation of RelA/p65 and has anti-inflammatory and anti-cancer effects [77]. However, due to its broad and non-specific targeting, it was not developed for clinical use. The development of two recent drugs that focus on the specific inhibition of IKKβ as a therapeutic approach for OC showed promising results in vitro and in vivo. Studies investigated the efficacy of the inhibitors IMD-0560 and IMD-0354 which are selective for IKKβ and prevent the phosphorylation of IκB, thereby sequestering NF-κB subunits in the cytoplasm to prevent transcriptional activity. Both effectively reduced p65 phosphorylation in OC cell lines with constitutive NF-κB activity, and treatment of OC cells with IMD-0560 drug reduced proliferation by inducing a G0/G1 cell cycle arrest [80,81]. Interestingly, a similar cell cycle effect was observed in OC cells with depleted IKKε, which caused G2/M arrest through p21 activation [82]. Inhibiting IKKβ with either drug also reduced VEGF production, endothelial tube formation and migration, showing potential for anti-angiogenic activity. Importantly, both drugs reduced tumor formation in vivo, suggesting potential activity against the stem-like population of OC cells [80,81]. Both drugs are under clinical development: a Phase 1 clinical trial was completed for IMD-0560 in Rheumatoid arthritis, Rheumatic osteoporosis and osteoarthritis; IMD-0354 completed proof-of-concept studies for non-cancer indications and is still under pre-clinical development for cancer indications [83].

Targeted inhibition of the regulatory IKKγ subunit—also known as NF-κB essential modulator (NEMO)—may provide an approach to selectively inhibit canonical NF-κB signaling. NEMO is required for the stability and activity of the IKKα/β complex that activates canonical signaling [84]. A novel bicyclic peptidyl inhibitor was developed against NEMO that demonstrated reasonable stability, cell permeability and potency against IKKβ phosphorylation in vitro [85]. This inhibitor, described as ‘peptide 7′, also affected the cell viability of OC cell lines in a dose-dependent manner without affecting the viability of a normal human ovarian surface epithelial cell line OSE at the same concentration [85]. Further development of peptide 7 as a drug for preclinical study is anticipated.

IKKε also has potential as a therapeutic target in OC, where it is elevated in metastatic OC and promotes adhesion and invasion of OC cell lines [14]. Silencing of IKKε in OC cell lines decreased the aggressiveness of tumors in vivo [14] and chemical inhibition of IKKε caused a G2/M arrest in vitro [82]. Inhibition of IKKε alone did not dramatically reduce proliferation of cell lines in vitro, but when combined with CHEK1 silencing or inhibition, achieved a significant reduction in proliferation and increased DNA damage and apoptosis [82]. This approach showed that IKKε inhibition combined with CHEK1 inhibition creates a ‘synthetic lethality’ that could be exploited in OC regardless of p53 status [82]. Unfortunately, no specific inhibitors for IKKε are approved for, or are under clinical development for cancer indications at present. Amlexanox is an inhibitor of IKKε and another NF-κB activator TANK-binding kinase 1 (TBK1), that was previously FDA approved for the treatment of apthous ulcers that has been investigated in a Phase 2 trial for the treatment of type 2 diabetes, insulin resistance, obesity and non-alcoholic fatty liver disease [86]. There is undoubtedly a potential anti-tumor benefit to targeting IKKs in OC cells, but blocking NF-κB activity too broadly may affect infiltrating immune cells such as macrophages and dendritic cells, which could render the intended therapy ineffectual by switching immune effectors towards an immunosuppressive phenotype [28].

### 4.2. Toll-Like Receptors and Myeloid Differentiation Primary Response 88 (Myd-88)

TLR signaling has both pro- and anti-tumor effects, making it a difficult pathway to target in ovarian cancer. TLRs 2,4 and the adaptor protein Myd-88 normally function in innate immune responses to pathogens and activate NF-κB signaling and produce inflammatory cytokines to mount an immune response [87]. However, TLR4 and Myd-88 are expressed in some cancer types including OC, where they correlate with poor prognosis [88,89]. TLR4 is implicated in mediating resistance to paclitaxel in OC cells, by activating NF-κB to induce expression of the multidrug resistance efflux pump *ABCB1*, and in the production of IL-6 and IL-8 cytokines [23,90]. Targeting TLR4 has been investigated as a therapeutic approach to sensitize OC cells to chemotherapy, inhibit migration and invasion and reduce NF-κB activity in OC cells [91,92]. Alternatively, stimulating TLR2 and TLR4 signaling can promote an anti-tumor immune response in OC models in combination with platinum therapy [93]. For example, a protein aggregate compound magnesium–ammonium phospholinoleate–palmitoleate anhydride (P-MAPA) was able to stimulate NF-κB activity in OC tissues, as assessed by p65 nuclear translocation, and increased TLR2 and TLR4 expression [93]. The expression of immune-modulating cytokines IL-6, TNFα and interferon gamma (IFNγ) was not significantly altered by the combination of cisplatin and P-MAPA, yet P-MAPA alone increased IFNγ expression in the OC tissues which can elicit anti-tumor immune responses by activation of innate immunity [93,94]. Future investigations of this pathway should delve into uncovering cancer-specific nodes in order to develop effective therapeutic strategies.

### 4.3. Lysophosphatidic Acid (LPA)

Lysophosphatidic acid (LPA) is abundant in the OC microenvironment and is known to promote invasion, angiogenesis and metastasis [95]. LPA stimulation increases the expression of VEGFR2 and cytokines VEGF, IL-8, TNFα and induces the expression of extracellular-matrix-degrading proteases in an NF-κB-dependent manner [95,96]. Inhibition of NF-κB activity in OC cell lines reduced their LPA-induced invasiveness by the reduced expression of MMP-2 and MMP-9 [95,96]. Additionally, LPA can activate NF-κB through the Ras-Rho-ROCK signaling cascade, and that specific inhibition of ROCK decreased the phosphorylation of IκBα, which inhibited NF-κB activity and inhibited the expression of the transcriptional target proteases [96].

### 4.4. microRNAs

The role of microRNAs (miRNAs) in mediating OC progression, metastasis and chemoresistance has been investigated with increasing importance in OC. A number of miRNAs have been reported to drive progression via or be suppressed by NF-κB signaling in OC (Table 1).

### 4.5. Targeted NF-κB Activation for Anti-Tumor Immunity

An alternative approach to promoting anti-tumor responses in OC is by targeting the TAMs and DCs to either deplete their numbers, or re-polarizing them to anti-cancer phenotypes. Activation of classical NF-κB signaling in TAMs and DCs to promote the production of inflammatory cytokines and innate immunity is one approach being investigated [104]. TAMs could be re-polarized from immunosuppressive to immunostimulatory by activating NF-κB, using specific delivery of siRNA to macrophages in a preclinical model of OC [105]. The siRNA targeted IκBα to reduce the inhibitory proteins that sequester NF-κB proteins in the cytoplasm and was delivered specifically to macrophages expressing the mannose receptor CD206, typically associated with immunosuppressive TAMs [105,106]. The siRNA was effectively delivered via mannosylated nanoparticles for targeted uptake by mannose-receptor-expressing macrophages, which then became cytotoxic to tumor cells in vitro and expressed an altered, inflammatory cytokine profile [105]. TLR agonists are another strategy to target NF-κB activation of canonical NF-κB activity in DCs and macrophages [104].

Motolimod is a TLR-8 agonist that was developed to target and activate monocytes and DCs and generate anti-tumor activity, which showed promise in a humanized mouse model of OC cell line xenografts [107]. Motolimod effectively activated human monocytes in a mouse model and increased the expression of inflammatory cytokines including TNFα [107]. Combined motolimod and pegylated liposomal doxorubicin (PLD) promoted immune-mediated cell death in HLA-matched OC cell xenografts in a pre-clinical model [107]. The combined therapy was able to efficiently reduce tumor burden and the therapy was progressed to a Phase 1 clinical trial to assess toxicity and dosage [107]. The combination therapy was further assessed in a Phase 2 trial for women with recurrent epithelial OC, but disappointingly, motolimod did not improve overall survival or progression-free survival compared to placebo [108]. The investigators did note, however, that patients treated with motolimod who demonstrated a local inflammatory response at the injection site had longer survival than motolimod-treated patients with no inflammatory response and suggested this may be useful as an indicator of patient’s tumor microenvironment and responsiveness to immune activation with motolimod [108].

## 5. Conclusions

The NF-κB signaling pathway has been implicated in OC tumorigenesis through its roles in mediating chemoresistance, maintenance of CSC populations, metastasis and immune evasion. The NF-κB signaling proteins and IKKs are overexpressed in OC tumors and correlate with poorer patient outcomes. Both canonical and non-canonical NF-κB signaling contribute to OC progression with overlapping and distinct roles in promoting chemoresistance and CSC maintenance. The inflammatory tumor microenvironment is also heavily influenced by NF-κB signaling, from the inflammatory cytokines that activate NF-κB signaling in OC cells and the immunosuppressive phenotype of TAMs and other immune cells that have diminished NF-κB activity. Targeting NF-κB in OC to eliminate the CSC population is desirable, but current strategies are limited to inhibition of IKKs. More research is needed to dissect the pathways and identify strategies to modulate NF-κB to reduce metastasis and tumor recurrence. There is a need to strike a balance between the inhibition of the non-canonical NF-κB in OC cells which enables long-term resistance and tumor re-initiation from CSC and the activation of canonical NF-κB in immune cells to generate an effective anti-tumor immune response.

## Figures and Tables

**Figure 1 cancers-11-01182-f001:**
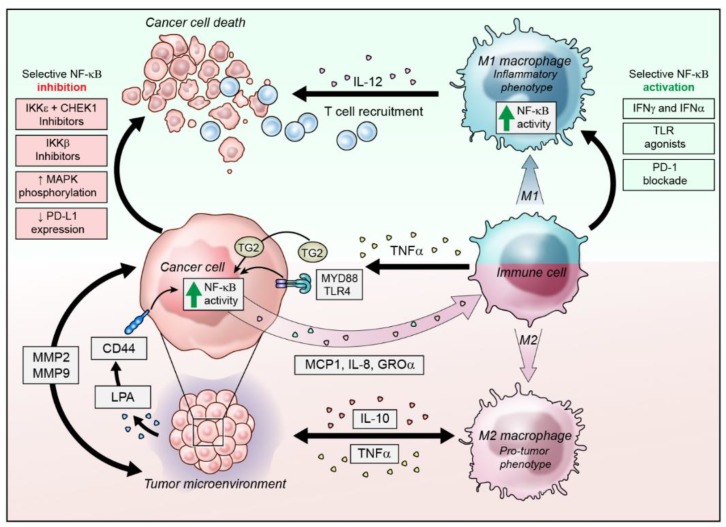
Strategies to target NF-κB in ovarian cancer. NF-κB signaling in cancer cells can be activated by interacting proteins in the tumor microenvironment, such as lysophosphatidic acid (LPA) signaling through CD44 to promote NF-κB activity and cytokine secretion. NF-κB signaling can also be activated by TNFα binding to cell surface receptors Myd-88 and TLR4, or through TNFα independent receptors such as TG2. NF-κB activity promotes cytokine secretion to dampen immune responses and switch macrophages from inflammatory M1 phenotypes to M2 immunoinhibitory phenotypes. Selective activation of NF-κB signaling in dendritic cells and macrophages with toll-like receptor (TLR) agonists or blockade of PD-1 could promote anti-tumor immunity and recruitment of additional anti-tumor immune cells. Alternatively, selective inhibition of NF-κB in tumor cells by IκB kinase (IKK) inhibitors could attenuate pro-tumor survival signaling in the tumor cells without disrupting inflammatory NF-κB activity in immune cells.

**Table 1 cancers-11-01182-t001:** miRNAs related to OC progression and NF-κB.

microRNA	Function Related to NF-κB Activity	References
miR-9	miR-9 targeted the 3’UTR of NF-κB to suppress NF-κB activation and MMP-9 expression leading to reduced invasion and cell growth.	[97,98]
miR-130α	miR-130a was activated by NF-κB signaling, miR-130a promoted proliferation and invasion of OC cells.	[99]
miR-503-5p	NF-κB-downregulated miR-503-5p and activated the JAK/STAT pathway. Inhibition of miR-503-5p by NF-κB enhanced invasion and migration in vitro.	[100]
miR-23α	miR-23a increased the expression of IKKα which promoted growth, migration, and invasion in OC cells in vitro	[101]
miR-134	TAB1 promoted pro-survival and anti-apoptosis via NF-κB signaling, TAB1 was inhibited by miR-134. NF-κB binds to the putative promoter region of miR-134 and represses its expression.	[102]
miR-141	miR-141 downregulates KEAP1 and activates NF-κB to mediate cisplatin resistance in vitro	[103]
miR-199a	miR-199a negatively regulates IKKβ expression in OC cells.	[69]

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
