# Peer review of "NF-κB Signaling in Ovarian Cancer"

_cancers, 2019, doi:10.3390/cancers11081182_

Round 1
Reviewer 1 Report
In this article, the authors review the de-regulation of the NF-κB signaling pathway in Ovarian cancer. Despite this subject has recently been discussed in excellent reviews, this is an ever-growing body of research and, therefore, it is a relevant topic for a novel review article.
I have only few suggestion to improve this manuscript:
1) The authors could add a novel chapter in which they discuss the NF-kB-dependent resistance to stressors of ovarian tumor cells (Kleinschmidt EG et al, Oncogene Oncogene. 2019 Jul 15).
2)In the context of NF-kB deregulation, the authors could discuss the impact of viral and microbial infections in the development of ovarian cancers.
Author Response
Comments and Suggestions for Authors
1) The authors could add a novel chapter in which they discuss the NF-kB-dependent resistance to stressors of ovarian tumor cells (Kleinschmidt EG et al, Oncogene. 2019 Jul 15).
We have incorporated the following text into section 2.1:
“NF-κB activity mediates resistance to stressors of OC cells that occurs during disease progression and metastasis such as oxidative stress and anchorage-independent survival [35]. Inhibition of NF-κB activity in cultured OC cell lines significantly inhibited anchorage-independent growth [35]. Canonical NF-κB signaling regulates the expression of several antioxidant response genes to alleviate elevated cellular reactive oxygen species levels in OC cells [35].”
2)In the context of NF-kB deregulation, the authors could discuss the impact of viral and microbial infections in the development of ovarian cancers.
We have incorporated the following text into section 3.2:
“Persistent inflammation has been associated with ovarian cancer development, in women with pelvic inflammatory disease or in response to pathogenic infections of Chlamydia trachomatis or Mycoplasma genitalium [66]. Recognition of pathogen-associated molecular patterns (PAMPs) by cell receptors such as TLRs initiates the innate immune response by activating NF-κB which stimulates the production of pro-inflammatory cytokines including TNFα, IL-6, IL-8, and MCP-1 [67]. Persistent, chronic inflammation following infection and constitutive NF-κB activity may contribute to ovarian carcinogenesis by producing pro-inflammatory and pro-angiogenic cytokines and through anti-apoptotic signaling [66].”
Reviewer 2 Report
The review by Harrington and Annunziata is interesting as it focuses in a specific aspect of NFkB signaling in ovarian cancer. The manuscript is relatively well written but could be improved by carefully revising some fragments of the text and paying more attention to the conclusion taken. Some examples:
Authors should be more cautious when mentioning BAY11-7082 as an IKK inhibitor since there are publications indicating that it is a general inhibitor of the ubiquitin system (Strickson S. BJ 2013).
In page 2 author include this categorical statement: “The prognostic significance of NF-κB activity in OC is related to its roles in promoting the chemoresistant and cancer stem cell populations that promote OC progression.” In my opinion this generalization is excessive and should be both attenuated and referenced.
Still in page 2, it is difficult to interpret (for non-NFkB experts) what “p105→p50 (NF-κB1), and p100→p52 (NF-κB2)” means.
“Degradation of p100” has to be changed to “partial degradation of p100 into p52, which take place in the proteasome”
Also when authors mention “NFkB expression” they have to identify the specific elements that are expressed or whether they refer to NFkB activity.
Also, several sentences in the manuscript need to be reformulated for clarity. Some examples are here shown but the whole manuscript should be revised to improve precision.
“There are strategies under development, however, that balance the targeted inhibition of NF-κB_in the tumor reinitiating cancer cell population and activation of the canonical NF-κB pathway in immune cells to promote anti-tumor immunity, which have led to some promising pre-clinical outcomes”
“IKKs have been investigated as a target for inhibiting NF-κB signaling due to their critical function allowing translocation for transcription to be activated.”
“Both pro and anti-apoptotic signaling that can be activated by NF-κB in OC cell lines, suggesting a biphasic role for NF-κB in OC [41].”
In page 3 all the text related with reference 41 has to be re-written to permit the readers to understand the results presented in this work.
When referring to IKKalpha i.e “The importance of IKKa in this model suggests that the CD44+ CSC population was maintained by the non-canonical NF-κB signaling pathway via RelB” it should be mentioned the possibility that the kinase is doing other pro-tumorigenic functions in addition to regulate RelB/p52 (i.e. Karin M, Miele L, Espinosa L publications)
“ Blockade of PD-1 on the DCs lead to activation of canonical NF-κB by degradation of IκBα for release of RelA/p65 for nuclear translocation and restored their pro-inflammatory functions [60].
A more general consideration is the absence of some pioneer articles that link the effect of damaging agents on NFkB activation to induce pro-survival activity in cells that should be included in the paper (i.e. the work from Miyamoto S and others).
Author Response
Reviewer 2
Comments and Suggestions for Authors
The review by Harrington and Annunziata is interesting as it focuses in a specific aspect of NFkB signaling in ovarian cancer. The manuscript is relatively well written but could be improved by carefully revising some fragments of the text and paying more attention to the conclusion taken. Some examples:
Authors should be more cautious when mentioning BAY11-7082 as an IKK inhibitor since there are publications indicating that it is a general inhibitor of the ubiquitin system (Strickson S. BJ 2013).The issue of BAY 11-7082 as a broad inhibitor is addressed in section 4.1. For accuracy, the text in section 2.3 referring to BAY 11-7082 as an IKK inhibitor has been changed to read as follows:
“…BAY 11-7082, an IKK inhibitor that prevents phosphorylation and degradation of IκBα ”
“…BAY 11-7082, an inhibitor with broad activity against the proteasome and ubiquitin system used to inhibit IKK, preventing phosphorylation and degradation of IκBα ”
In page 2 author include this categorical statement: “The prognostic significance of NF-κB activity in OC is related to its roles in promoting the chemoresistant and cancer stem cell populations that promote OC progression.” In my opinion this generalization is excessive and should be both attenuated and referenced.The text has been changed to read as follows:
“The prognostic significance of NF-κB activity in OC is related to its roles in promoting properties of aggressive OC such as chemoresistance [15,20-24] and maintenance of cancer stem cell populations that contribute to recurrence [3,25].”
Still in page 2, it is difficult to interpret (for non-NFkB experts) what “p105→p50 (NF-κB1), and p100→p52 (NF-κB2)” means.The text has been changed to read as follows:
“NF-κB is a group of transcription factors consisting of 5 proteins; RelA/p65, c-Rel, RelB, p50 (NF-κB 1) and p52 (NF-κB 2) which are generated by selective degradation in the proteasome of p105 and p100 respectively [22,23].”
“Degradation of p100” has to be changed to “partial degradation of p100 into p52, which take place in the proteasome”The text has been changed to read as follows:
“..and partial degradation of p100 which generates p52 in the proteasome..”
Also when authors mention “NFkB expression” they have to identify the specific elements that are expressed or whether they refer to NFkB activity.
We reviewed the manuscript and found only one example where “NF-κB expression” was written, in the second paragraph of the introduction. The text has been changed as follows:
“In OC, the expression levels of NF-κB proteins p65 and p50 are elevated in malignant and borderline serous ovarian tumors…”
Also, several sentences in the manuscript need to be reformulated for clarity. Some examples are here shown but the whole manuscript should be revised to improve precision.
We have reviewed the manuscript text and made typographical changes and restructured sentences for improved clarity. These changes are indicated with the track changes feature in the resubmitted document.
Section 2.2 text has been changed for clarity:
“In CD44+ SKOV3 cells there was increased expression of RelA, but there was greater non-canonical IKKα and RelB as well as stem cell markers NANOG, SOX2 and OCT4/3 compared to CD44- cells, thus identifying the CSC population”
In CD44+ SKOV3 cells there was increased expression of RelA but the expression of non-canonical NF-κB proteins RelB and IKKα was greater, as was the expression of stem cell markers NANOG, SOX2 and OCT4/3 compared to CD44- cells.
Section 3.1 text has been changed for clarity:
“In a co-culture experiment of mouse bone marrow derived macrophages (BMDMs) with mouse ovarian cancer cell line ID-8, specific knockout of upstream activators of canonical NF-κB signaling MyD88, IL1R, TLR2 or TLR4 in BMDMs decreased the invasion of ID-8 cells in vitro and tumor growth in vivo”
In a co-culture experiment of mouse bone marrow derived macrophages (BMDMs) with mouse ovarian cancer cell line ID-8, the importance of upstream activators of canonical NF-κB signaling was demonstrated by specific knockouts of Myd-88, IL1R, TLR2 and TLR4 in the BMDMs which caused decreased invasion of ID-8 cells in vitro and decreased tumor growth in vivo.
“A target for cancer immunotherapy is Programmed cell death-1 (PD-1), which is a negative regulator expressed on immune cells, particularly T cells, that is directed by interaction with its ligand (PD-L1) which is expressed on normal cells but also cancer cells to avoid immune cytotoxicity”
“A target for cancer immunotherapy is programmed cell death-1 (PD-1), which is a negative regulator of immune cytotoxicity that is expressed on immune cells, particularly T cells. Regulation of immune cytotoxicity through PD-1 is directed by interaction with its ligand (PD-L1), which is expressed on normal cells but has also been detected on cancer cells as a strategy to avoid immune cytotoxicity.”
“There are strategies under development, however, that balance the targeted inhibition of NF-κB_in the tumor reinitiating cancer cell population and activation of the canonical NF-κB pathway in immune cells to promote anti-tumor immunity, which have led to some promising pre-clinical outcomes”
The text has been changed to read as follows:
“There are some promising pre-clinical outcomes from developing research to target and inhibit NF-κB only in the tumor-reinitiating cancer cell population of OC and concurrently activate canonical NF-κB signaling in immune cells to promote anti-tumor immunity.”
“IKKs have been investigated as a target for inhibiting NF-κB signaling due to their critical function allowing translocation for transcription to be activated.”
The text has been changed to read as follows:
“Due to their critical function in the activation of NF-κB signaling, IKKs have been investigated as a target for inhibitors to prevent the nuclear translocation of NF-κB subunits for transcriptional activity.”
“Both pro and anti-apoptotic signaling that can be activated by NF-κB in OC cell lines, suggesting a biphasic role for NF-κB in OC [41].”
See below text changes.
In page 3 all the text related with reference 41 has to be re-written to permit the readers to understand the results presented in this work.
To address this we have restructured the text related to this reference to better explain the reference regarding the potential for pro and anti-apoptotic signaling in ovarian cancer. The text has been changed to read as follows:
“Both pro and anti-apoptotic signaling that can be activated downstream of NF-κB in OC cell lines, suggesting a biphasic role for NF-κB in OC progression [42]. In a comparison of chemoresistant cell lines with high NF-κB transcriptional activity with their isogenic, chemosensitive parental cells it was found that inhibition of the NF-κB pathway using a dominant negative phosphorylation-deficient mutant of IκBα had opposing effects on the chemoresistant and chemosensitive cells of the same origin. [42]. The IκBα mutation caused reduced tumour growth of the aggressive, chemoresistant cells in vivo but this same mutation promoted tumor growth in the isogenic chemosensitive cells. Apoptosis could be induced in the aggressive chemoresistant cell lines by introduction of this mutation whereas the chemosensitive cells showed reduced apoptosis [42]. This polar difference in activity and cell viability was related to the ability of NF-κB to differentially regulate mitogen-activated protein kinase (MAPK) phosphorylation to produce pro- or anti- apoptotic signaling in OC cells [42]. This study highlighted the canonical pathway activity and its potential to contribute to chemoresistance and OC cell survival.”
When referring to IKKalpha i.e “The importance of IKKa in this model suggests that the CD44+ CSC population was maintained by the non-canonical NF-κB signaling pathway via RelB” it should be mentioned the possibility that the kinase is doing other pro-tumorigenic functions in addition to regulate RelB/p52 (i.e. Karin M, Miele L, Espinosa L publications).
To address this, the following text in section 2.2 has been added to and changed:
“While it should be noted that IKKα can be activated and function separately from NF-κB pathway signaling in other cancer types , no direct evidence of this has been shown in OC as yet. Thus the importance of IKKα in this model suggests that the CD44+ CSC population was likely maintained by the non-canonical NF-κB signaling pathway via RelB.”
“Blockade of PD-1 on the DCs lead to activation of canonical NF-κB by degradation of IκBα for release of RelA/p65 for nuclear translocation and restored their pro-inflammatory functions [60].
The text has been changed to read as follows:
“Blockade of PD-1 on the DCs restored their pro-inflammatory functions by activation of canonical NF-κB with degradation of IκBα allowing the nuclear translocation of RelA/p65”
A more general consideration is the absence of some pioneer articles that link the effect of damaging agents on NFkB activation to induce pro-survival activity in cells that should be included in the paper (i.e. the work from Miyamoto S and others).
To address this, the following text in section 2.1 has been added:
“DNA damage can also activate NF-κB signaling in a retrograde nuclear to cytoplasmic signaling cascade that is dependent on ataxia telangiectasia-mutated (ATM) [42]. Chemotherapeutic agents that cause genotoxic stress such as etoposide and doxorubicin activate NF-κB pro-survival signaling [43,44]. In response to DNA damage IKKγ subunit also known as NF-κB essential modulator (NEMO) translocates into the nucleus for post-translational modification, phosphorylation by ATM and monoubiquitination for nuclear export [42,45]. In the cytoplasm, the modified NEMO-ATM complex can activate IKKs to phosphorylate and degrade inhibitory IκBα subunits and activate canonical NF-κB survival signaling [42].”